# Cytotoxicity and Autophagy Induced by Ivermectin via AMPK/mTOR Signaling Pathway in RAW264.7 Cells

**DOI:** 10.3390/molecules28052201

**Published:** 2023-02-27

**Authors:** Xiang Wang, Jian Wang, Ping Zhang, Cheng Zhang, Weiguo Wang, Mengqi Wu, Wenping Xu, Liming Tao, Zhong Li, Yang Zhang

**Affiliations:** 1Shanghai Key Laboratory of Chemical Biology, School of Pharmacy, East China University of Science and Technology, Shanghai 200237, China; 2Department of Imaging, Weifang Hospital of Traditional Chinese Medicine, Shandong 261041, China; 3Department of Pathology, UT Southwestern Medical Center, Dallas, TX 75390, USA

**Keywords:** ivermectin, cytotoxicity, cell cycle arrest, autophagy, AMPK/mTOR pathway

## Abstract

The widespread and excessive use of ivermectin (IVM) will not only cause serious environmental pollution, but will also affect metabolism of humans and other mammals that are exposed. IVM has the characteristics of being widely distributed and slowly metabolized, which will cause potential toxicity to the body. We focused on the metabolic pathway and mechanism of toxicity of IVM on RAW264.7 cells. Colony formation and LDH detection assay showed that IVM significantly inhibited the proliferation of and induced cytotoxicity in RAW264.7 cells. Intracellular biochemical analysis using Western blotting assay showed that LC3-B and Beclin-1 were upregulated and p62 was down-regulated. The combination of confocal fluorescence, calcein-AM/CoCl_2_, and fluorescence probe results showed that IVM could induce the opening of the mitochondrial membrane permeability transition pore, reduce mitochondrial content, and increase lysosome content. In addition, we focused on induction of IVM in the autophagy signal pathway. The Western blotting results showed that IVM increased expression of p-AMPK and decreased p-mTOR and p-S6K expression in protein levels, indicating that IVM activated the AMPK/mTOR signaling pathway. Therefore, IVM may inhibit cell proliferation by inducing cell cycle arrest and autophagy.

## 1. Introduction

Ivermectin (IVM) is a highly liposoluble drug with the characteristics of being widely distributed and slowly metabolized and eliminated after entering the body [1]. Meanwhile, IVM can enter most organs and tissues in the body, including the stomach, intestines, lungs, skin, and adipose [2,3,4,5]. However, because of the existence of the blood–brain barrier, it is believed that IVM is safe for mammals. In recent years, IVM has been widely used in agriculture, veterinary medicine, and aquaculture. The extensive use of IVM has caused extensive environmental exposure, posing a health risk to a variety of non-targeted organisms [6,7,8]. A field study in a floodplain wetland showed that the total concentration of IVM in the environment was 494.4 ng/g, and it could accumulate during lactation through the biological amplification of trophic webs [9]. In addition, IVM has been reported to be toxic to soil invertebrates [10] and plants [11]. Therefore, the safety of IVM on mammals needs further research.

Autophagy (type II programmed cell death), a catabolic process strictly controlled by genes, is an important mechanism that cells use to cope with intracellular and extracellular stress [12,13,14]. Excessive autophagy can lead to autophagic death [15,16]. Autophagic vesicles appear early in the autophagy process. It has been established that autophagy may arise from various organelles in intramembranous systems, such as mitochondria. Selective autophagy of mitochondria, known as mitophagy, is an important mitochondrial quality control mechanism that eliminates damaged mitochondria [17].

Each stage of the autophagy process is strictly regulated by a limited number of highly conserved autophagy-associated proteins. Beclin-1 is the initiating protein of the autophagosome, while LC3 is the structural component of the autophagosome [18,19]. P62 is mammalian target of rapamycin complex 1 (mTORC1), which acts as a signaling hub to activate lysosomes [20]. One of the causes of P62 degradation is autophagy. Beclin-1, LC3, and p62 are three major autophagy-related proteins widely used in autophagy [21]. It has been proved that autophagy is regulated by multiple signaling pathways. The mammalian target of rapamycin (mTOR), as one of the most important proteins in autophagy, plays a crucial role in cell growth and proliferation [22]. AMP-activated protein kinase (AMPK), a serine–threonine protein kinase, is a crucial physiological energy sensor that regulates cell and organism energy homeostasis [23]. Meanwhile, AMPK can negatively regulate mTOR, thus inducing autophagy [24].

This study focuses on the immunotoxicity caused by IVM. The immune system is an important system widely distributed in the body and plays an indispensable role in host defense and homeostasis maintenance [25,26]. Interference with various components of the immune system may lead to excessive or insufficient vectors in the immune function of the body, thereby reducing immunity. Macrophages play an important role in removing foreign bodies, presenting antigens, and regulating inflammatory responses [27]. The survival ability of macrophages is the basis of immune function. When macrophages are damaged, it will cause damage to the immune system. Our previous studies had shown that IVM causes DNA damage in macrophages (RAW264.7 cells) and induced apoptosis, resulted in immunotoxicity associated with the NF-κB signaling pathway in RAW264.7 cells [28]. Therefore, it is of great significance to study immunotoxicity through macrophages. In this study, mouse macrophages (RAW264.7 cells) were used as the cell model to preliminarily evaluate the immunotoxicity of IVM according to its induced cytotoxicity and autophagy ability.

## 2. Results

### 2.1. Colony Formation and LDH Activity

Colony formation assay was used to evaluate the cytotoxicity of IVM on RAW264.7 cells. The results showed that IVM could significantly inhibit the proliferation of RAW264.7 cells (Figure 1A). Additionally, the percentage of colonies decreased after treatment with IVM in a concentration-dependent manner in RAW264.7 cells (Figure 1B). The LDH release continued to rise with treatment assay IVM concentration (Figure 1C). The LDH activity of 20 μM IVM was 2.5 times higher than that of the control group, and there was a significant difference between the two groups. Therefore, IVM induced cytotoxicity in RAW264.7 cells.

### 2.2. IVM Induced Cell Cycle Arrest

RAW264.7 cells were treated with IVM for 24 h, and the distribution of the cell cycle was studied. The results showed that the percentage of G1-phase cells was significantly increased in a concentration-dependent manner (Figure 2A,B). Meanwhile, the percentage of S-phase and G2-phase cells decreased gradually compared with the control group. To further clarify the effect of IVM on the RAW264.7 cell cycle, we detected the expression of cell cycle-related proteins. The G1-phase proteins cyclin D1 and CDK4 gradually decreased with the increase of concentration (Figure 2C). Therefore, we found that IVM induced cell cycle arrest in RAW264.7 cells.

### 2.3. IVM Induced Autophagy

MDC is an eosinophilic fluorescent probe, which is a specific dye for detecting autophagy. It preferentially accumulates on autophagic vesicles to form a fluorescent focus. The fluorescence intensity of MDC was detected and analyzed by a fluorescence microscope to reflect whether the cells formed autophagic vesicles. The results showed that the fluorescence focus of RAW264.7 cells increased gradually, indicating that the number of autophagosomes increased significantly (Figure 3A). In order to further explore the effect of IVM on autophagy in RAW264.7 cells, we detected the expression of autophagy-related proteins. The protein expression of Beclin-1 and LC3-B increased, while the protein expression of p62 decreased (Figure 3B,C). Therefore, IVM induced autophagy in RAW264.7 cells.

### 2.4. IVM Induced Mitochondrial Damage and Mitophagy

The calcein-AM/CoCl_2_ fluorescence-quenching experiment was used to detect the opening of the mitochondrial permeability transition pore (mPTP) to evaluate the change in mitochondrial membrane permeability. Calcein-AM was loaded onto the cells of the control group, and the fluorescence could be seen throughout the whole cell (Figure 4A). The fluorescence of calcein-AM was quenched in the presence of CO^2+^. When IVM concentration increased, mitochondrial fluorescence first increased and then decreased, indicating the opening of the mPTP channel and the change in mitochondrial membrane permeability (Figure 4A). ATP is the main source of energy for the cell, and electron transfer in the mitochondrial oxidative respiratory chain provides the necessary energy drive for ATP synthesis. The integrity of mitochondrial structure and function is closely related to changes in the concentration of ATP in the cell. The intracellular ATP content was detected with an ATP assay kit, and the results showed that ATP content decreased significantly with the increase in IVM concentration (Figure 4B). Damaged mitochondria were encapsulated by lysosomes, which form autophagic lysosomes for degradation. Therefore, loss of mitochondrial structure and function leads to increased levels of autophagosomes in mitochondria. When the concentration of IVM increased (from 0 μM to 20 μM), the MitoTracker Green stain for mitochondrial content decreased gradually while the lysosomal content represented by LysoTracker Red increased gradually (Figure 5). In summary, IVM induced mitochondrial damage and mitophagy in RAW264.7 cells.

### 2.5. IVM Activated the AMPK/mTOR Pathway

Decreased ATP content can activate the AMPK/mTOR autophagy-signaling pathway, which leads to autophagy. The phosphorylation levels of AMPK, mTOR, and p70s6k were detected by Western blot (Figure 6A). The results showed that p-AMPK increased and p-mTOR and p-p70s6k decreased in RAW264.7 cells induced by IVM, and the expression level showed a concentration-dependent manner (Figure 6B). Therefore, IVM induced autophagy in RAW264.7 cells by activating the AMPK/mTOR pathway.

## 3. Discussion

IVM is a highly effective anti-parasitic veterinary drug, and its extensive use causes risks to the health of mammals. With the wide application of synthetic genetic algorithms [29], the security of IVM has received increasing attention. The European Medicines Agency has established maximum residue limits (MRL) of 100 μg/kg for IVM in fat and liver and 30 μg/kg in kidney for all mammalian food products [30]. However, IVM has not established MRL in mammalian immune tissue. Strong lipid solubility leads to the spread of IVM in many mammalian tissues after administration. The mammalian immune system is spread throughout the body and includes a variety of cells, organs, proteins, and tissues [31]. A compromised or injured immune system reduces the body’s resistance to other pathogens and increases the body’s susceptibility to various diseases. After exposure to exogenous substances, the immune system may exhibit immunotoxicity before other organs. However, few studies have been conducted on the effects of IVM on the immune system.

We focused on the immunocytotoxicity of IVM in mammals after exposure. We confirmed that IVM inhibited the proliferation of RAW264.7 cells by colony formation assay, and the high LDH activity is a sign of plasma membrane damage and indicates necrotic cell death [32]. Meanwhile, IVM induced the down-regulation of cyclin D1 and CDK4, and caused cell cycle arrest. The results showed that IVM had obvious cytotoxicity to RAW264.7 cells. The main forms of cell death include apoptosis, necrosis, and autophagy [33]. Previous studies had shown that avermectin (AVM), a macrolide insecticide, can induce autophagic responses in a variety of cells and tissues [34,35,36]. Subsequently, researchers including our group found that IVM, as a derivative of AVM, exhibits similar biological functions in non-target mammalian cells and can also induce cellular autophagy or apoptosis [28,37,38,39,40]. Autophagy and apoptosis constitute functionally distinct mechanisms for the turnover or destruction of cytoplasmic structures within cells and of cells within organisms, respectively [41]. Therefore, in this study, we focused our attention on the mechanism of autophagy. Notably, the results showed that IVM induced autophagic vesicles in RAW264.7 cells.

Autophagy can consist of four stages. The first stage (initial stage): after receiving the autophagy induction signal, the cell forms a double layer of the membrane in the cytoplasm, and then extends to the sides, thus forming a double bowl-like structure. This stage marks the beginning of autophagy and requires the participation of Beclin-1 [42]. The second stage (expansion): the autophagy precursor continues to expand and extend, wrapping the material that needs to be degraded in a bubble, forming a closed spherical autophagosome. This stage is the extension of autophagic vesicles and requires the participation of LC3 protein. Lc3-b is distributed in autophagic vesicles and autophagosomes and is a biomarker of autophagy [43]. The third stage (maturity): after their formation, autophagosomes fuse with lysosomes to form autophagolysosomes. The fourth stage (degradation): autophagic lysosomes contain a variety of enzymes. Autophagic lysosomes can be degraded, and the released fatty acids, amino acids, and ATP are transported into the cytoplasm [44]. Both maturation and degradation require the participation of P62 protein. In this study, Beclin-l down-regulation, LC3-B production, and P62 degradation were observed, confirming that autophagy occurred at the protein level.

In addition, we focused on mitochondrial autophagy. Mitochondria are the main source of energy and play an important role in regulating cell metabolism [45]. Mitochondria, one of the most sensitive organelles, are easily destroyed by physical and chemical factors in the environment. To dispose of these damaged mitochondria quickly, cells undergo mitochondrial autophagy [46]. The co-localization of mitochondria and lysosomes was observed, which showed that the content of mitochondria decreased and the content of lysosomes increased. Therefore, the autophagic type of the RAW264.7 cells induced by IVM was mitochondrial autophagy.

This study focused on the mechanism of autophagy. Autophagy is regulated by a series of signaling molecules, in which multiple signal transduction pathways converge on mTOR, suggesting that mTOR plays a key role in autophagy [47]. AMP-activated protein kinase (AMPK), as an important sensor for maintaining cellular energy production, can be activated by intracellular ATP consumption [48]. The results showed that up-regulation of AMPK phosphorylation and down-regulation of mTOR and 70S6K phosphorylation were concentration-dependent. Therefore, IVM mediated mitochondrial autophagy through the AMPK/mTOR signaling pathway.

## 4. Materials and Methods

### 4.1. Cell Culture and IVM Treatments

RAW264.7 macrophages (ATCC, TIB-71) were grown in T25 flasks under standard conditions (5% CO_2_, 37 °C) using DMEM medium (HyClone, Logan, UT, USA) with stable glutamine, supplemented with 10% *v/v* fetal bovine serum (FBS, Gibco, Carlsbad, CA, USA), 100 U/mL penicillin, and 100 μg/mL streptomycin (Gibco, Carlsbad, CA, USA). The cells were subcultured before the density approached 80–90% and the DMEM medium was changed every 2–3 days. Exponentially growing RAW264.7 cells were seeded at a density of 5 × 10^4^ cells/mL in 6-well culture plates or 60 mm culture dishes, depending on the experimental design.

Ivermectin (IVM, 22, 23-dihydro avermectins B1a, consisting of ≥98%, CAS: 70288-86-7) was obtained from Sigma-Aldrich (Sigma, St. Louis, MO, USA). IVM was dissolved in DMSO to form a solution of 20 mM which was diluted to a specific concentration with a culture medium. The cells were treated with IVM at concentrations of 2.5, 5, 10, and 20 μM, and the final concentration of DMSO was less than 0.1% *v*/*v*.

### 4.2. Colony Formation Assay

The effect of compounds on cell proliferation can be determined by colony formation assay [49]. RAW264.7 cells were treated with IVM for 6 h and then cultured in an incubator for 14 days. Fresh medium was replaced every 3 days during the incubation period. The cells were fixed with methanol (15 min) and stained with a 1:10 dilution of Giemsa (Sigma, St. Louis, MO, USA) reagent (10 min). Finally, the number of cell clones was counted under a microscope.

### 4.3. Lactate Dehydrogenase (LDH) Assay

The cytotoxicity of compounds was determined by detecting LDH activity measured with a LDH kit (Beyotime, Shanghai, China) according to the instruction manual. Centrifuge tubes were used to collect cell media from RAW264.7 cells after 6 h of IVM treatment. The medium was centrifuged for 15 min (4 °C, 2000 rpm) and LDH activity was measured.

### 4.4. Cell Cycle Assay

The effect of IVM on RAW264.7 cells was evaluated according to the previous study method by seeding cells in 60 mm culture dishes [50]. RAW264.7 cells were treated with IVM for 24 h and collected into centrifuge tubes. The cells were washed twice with PBS and then immobilized overnight with 70% frozen ethanol. Subsequently, the cells were washed twice with PBS and incubated with the PI solution in PBS supplemented with TritonX-100 (0.2%) and RNase in the dark at 37 °C for 30 min. Finally, the percentage of cells in G0/G1, S, and G2/M phases was calculated with a cell analyzer (Cytoflex, Beckman Coulter, Chaska, MN, USA).

### 4.5. Monodansylcadaverine (MDC) Staining

The autophagic vacuoles were labeled with MDC (Sigma, St. Louis, MO, USA) according to the previous method [51]. The cells were cultured in 10 mΜ MDC working solution in the dark at 37 °C (20 min) and then washed twice with PBS (pH 7.4). A fluorescence microscope (lessons, DM3000, GER) was used to take photos and analyze the treated cells.

### 4.6. Colocalization of Mitochondria and Lysosomes

MitoTracker Green was used to stain the mitochondria of living cells. LysoTracker Red was used to stain the lysosomes of living cells. The two probes were used for colocalization imaging of mitochondria and lysosomes, as previously described [51]. RAW264.7 cells were treated with IVM for 6 h, then co-cultured with MitoTracker Green (0.5 μM) and LysoTracker Red (0.5 μM) for 30 min, and photos were recorded with a fluorescence microscope (lessons, DM3000, GER).

### 4.7. Mitochondrial Permeability Transition Pore (mPTP) Assay

The opening of mPTP was determined by the calcein-AM and CoCl_2_ method [28]. RAW264.7 cells were seeded in 6-well plates and treated with IVM for 6 h. After washing with PBS (pH 7.4), the cells were incubated with 1 mM calcein-AM in an incubator (20 min). The cells were loaded with 1 mM CoCl_2_ for 30 min and the group without CoCl_2_ was used as the positive control. A fluorescence microscope (lessons, DM3000, GER) was used to take photos and analyze the treated cells.

### 4.8. Intracellular ATP Assay

Intracellular ATP levels were determined according to the method in the instruction manual of the ATP Kit (Beyotime, Shanghai, China). RAW264.7 cells were treated with IVM for 6 h and lysed in lysis buffer at 4 °C (20 min). After centrifugation at 12,000 rpm for 15 min, part of the supernatant was transferred to 96-well plates to determine ATP content, and the ATP content of the other part was determined with a BCA protein detection kit (Beyotime, Shanghai, China). ATP concentration (μM) was normalized to protein concentration (mg/L).

### 4.9. Western Blotting

The cells were collected and lysed in RIPA buffer containing PMSF (Sigma, St. Louis, MO, USA). A BCA protein detection kit was used to detect the total protein concentration. The same amount of protein was separated by 8–12% sodium alkyl sulfate–polyacrylamide gel electrophoresis and then transferred to the PVDF membrane. The membrane was sealed with a blocking buffer for 2 h at room temperature. Then, the membrane was cultured in blocking buffer at 4 °C overnight with a 1:1000 dilution of primary antibody and washed 3 times with TBST. Finally, the membrane was cultured with a secondary antibody in 1:4000 dilution for 1 h and washed 3 times with TBST. Protein bands were detected using the ECL kit (Yeasen, Shanghai, China) in a chemiluminescence gel imaging system (Tanon, Shanghai, China).

### 4.10. Statistical Analysis

At least 3 independent tests were performed for each analysis. The results showed mean ± SD. Dunnett’s test was performed using an analysis of variance. SPSS 26.0 software was used for statistical analysis. The difference was considered statistically significant, * *p* ≤ 0.05, ** *p* ≤ 0.01, *** *p* ≤ 0.001.

## 5. Conclusions

In conclusion, IVM significantly inhibited the proliferation of RAW264.7 cells and induced autophagy. Under special circumstances, autophagy helps to induce apoptosis or necrosis [41]. Combined with the previous results, IVM can induce apoptosis and DNA damage of RAW264.7 cells [28]. Therefore, IVM may inhibit cell proliferation through cell cycle arrest, autophagy, and apoptosis. The cytotoxicity of IVM to macrophages in vitro suggests that IVM may cause potential harm to the immune system. In future research, we will focus on the relationship between cell cycle arrest and autophagy, and explore other potential mechanisms of IVM-induced immunotoxicity.

## Figures and Tables

**Figure 1 molecules-28-02201-f001:**
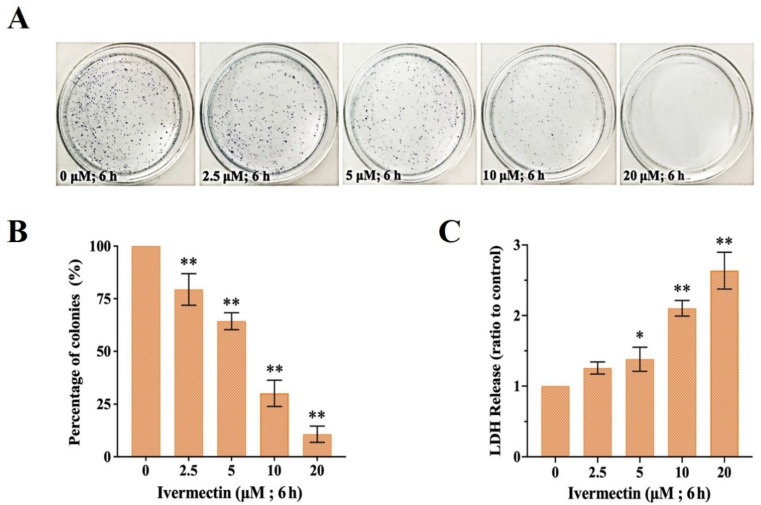
IVM induces cytotoxicity in RAW264.7 cells. (**A**) Colony formation of cells after IVM treatment of cells for 6 h. (**B**) The number of colonies was significantly decreased in cells treated with IVM for 6 h. (**C**) Release rate of LDH in the treatment of various concentrations of IVM for 6 h. Data were expressed as means ± SD (standard deviation) of three separate sets of independent experiments. * *p* < 0.05, ** *p* < 0.01 and indicate the significant differences from the control.

**Figure 2 molecules-28-02201-f002:**
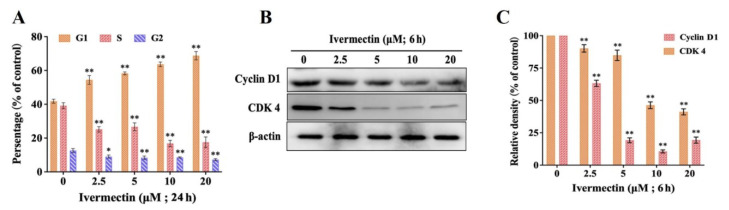
IVM induces cell cycle arrest in RAW264.7 cells. (**A**) Percentage of cells in G1-, G2-, and S-phases after IVM treatment of RAW264.7 cells for 24 h. (**B**) Cyclin D1 and CDK 4 expression in RAW264.7 cells treated with IVM for 6 h. (**C**) Densitometric analysis of the expression levels of cyclin D1 and CDK 4. Data were expressed as means ± SD (standard deviation) of three separate sets of independent experiments. * *p* < 0.05, ** *p* < 0.01, and indicate the significant differences from the control.

**Figure 3 molecules-28-02201-f003:**
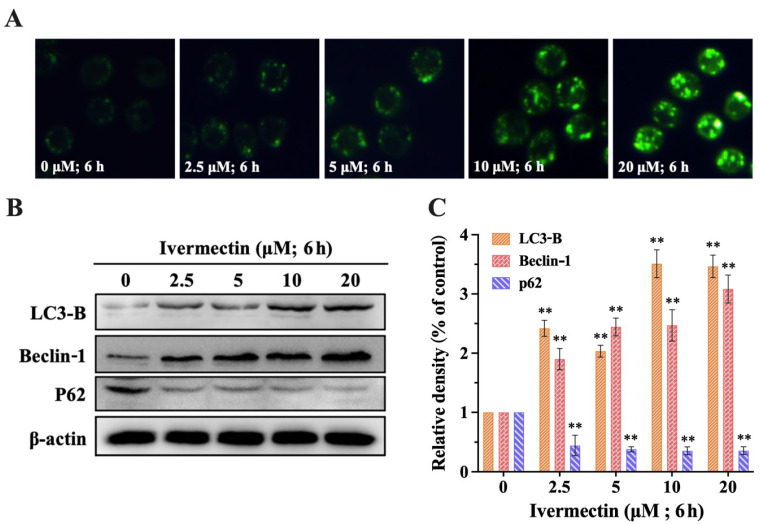
IVM induces autophagy in RAW264.7 cells. (**A**) Representative fluorescence photomicrograph (200×) of RAW264.7 cells stained with MDC after treatment with different concentrations for 6 h. (**B**) LC3-B, Beclin-1, and p62 expression in RAW264.7 cells treated with IVM for 6 h. (**C**) Densitometric analysis of the expression levels of LC3-B, Beclin-1, and p62. Data were expressed as means ± SD (standard deviation) of three separate sets of independent experiments. ** *p* < 0.01, and indicate the significant differences from the control.

**Figure 4 molecules-28-02201-f004:**
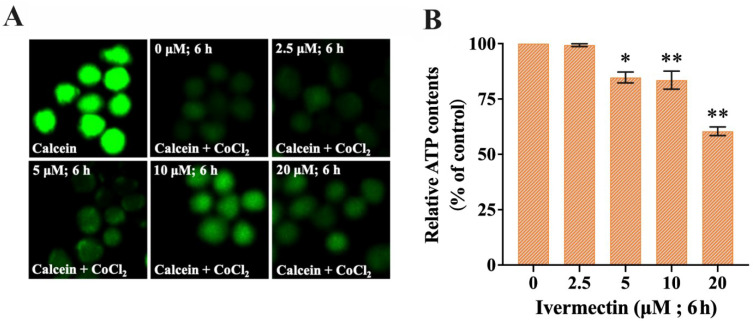
The effect of IVM on ATP content in RAW264.7 cells. (**A**) After IVM treatment of cells for 6 h, the mPTP opening was detected by co-loading with calcein-AM and CoCl_2_ (200×). (**B**) ATP content was measured after IVM treatment of cells for 6 h. Data were expressed as means ± SD (standard deviation) of three separate sets of independent experiments. * *p* < 0.05, ** *p* < 0.01, and indicate significant differences from the control.

**Figure 5 molecules-28-02201-f005:**
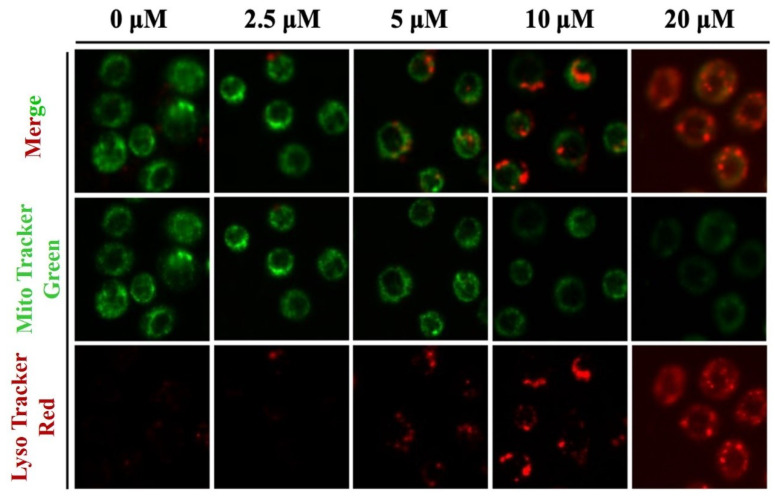
IVM induces mitophagy in RAW264.7 cells. After being treated with IVM for 6 h, cells were stained with MitoTracker Green and LysoTracker Red and observed with a fluorescence microscope (200×).

**Figure 6 molecules-28-02201-f006:**
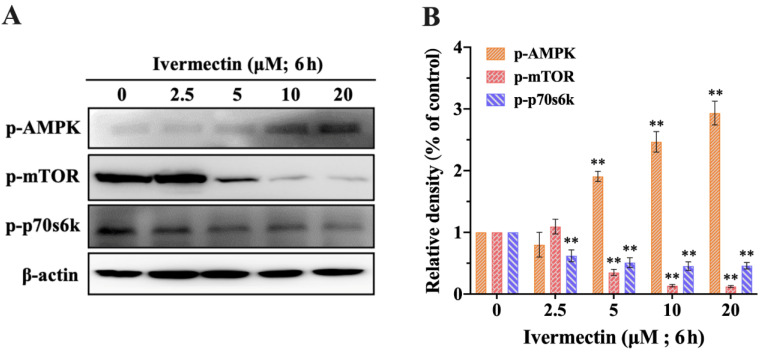
The effect of IVM on AMPK/mTOR signaling pathways in RAW264.7 cells. (**A**) Phosphorylated mTOR, AMPK, and p70s6k expression in RAW264.7 cells treated with IVM for 6 h. (**B**) Densitometric analysis of the expression levels of phosphorylated mTOR, AMPK, and p70s6k. Data were expressed as means ± SD (standard deviation) of three separate sets of independent experiments. ** *p* < 0.01 and indicate significant differences from the control.

## Data Availability

Not applicable.

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
