# Peer review of "Cytotoxicity and Autophagy Induced by Ivermectin via AMPK/mTOR Signaling Pathway in RAW264.7 Cells"

_molecules, 2023, doi:10.3390/molecules28052201_

Round 1
Reviewer 1 Report
Ivermectin (IVM) is a commercially well-known antiparasitic agent derived from the natural fermentation of avermectin. Originally used as a veterinary drug, IVM has been studied for its pharmacokinetic advantages, such as anticancer, antimigration, and antiproliferative effects, using several cell types in several publications. Authors of this manuscript focused on the cytotoxicity and pathway mechanism of Ivermectin (IVM) using RAW264.7 cells (macrophages). The analyzes performed are of good quality and clearly demonstrate the toxicity of IVM to cells. Ultimately, they come to similar conclusions as other authors, that this compound is toxic and causes a number of effects such as autophagy, mitophagy and ultimately leads to apoptosis
The quality of the presented results is good with the exception of Fig.1A
Cytometric analysis is illegible and needs to be corrected
Author Response
Dear Reviewer
Thanks for your comments concerning our manuscript entitled “Cytotoxicity and autophagy induced by Ivermectin via AMPK/mTOR signaling pathway in RAW264.7 cells” (Manuscript ID: molecules-2167456). Those comments are all valuable and helpful for revising and improving our paper. We have studied all comments carefully and have made conscientious correction. The main corrections in the paper and the responds to your comments are as following:
Comment 1: Cytometric analysis is illegible and needs to be corrected
Response: Thanks for your advice. We have adjusted the contrast and sharpness of the Fig.1A to make the image clearer, and the old picture has been replaced.

Reviewer 2 Report
The article is interesting and well done. However it has some limitations that should be highlighted. For example, in introduction or in discussion, it must be mentioned that this work is a continuation of their research, since the same group had already shown that ivermectin induces the death of RAW cells (Zhang P 2022 Chemosphere, 289), although in this article they do not yet describe the type of death. It is necessary also to discuss that previously has been demostrated that ivermectin could induce autophagy in other cell line. Another limitation of the work, that must be declared, is that RAW cells are tumor cells, and it has to be compared the effect in normal cells,even between resting macrophages, M1 and M2 . Another limitation is the concentration of ivermectin tested, the minimun was 2.5 uM that is around 2 ug/mL.
By another hand, figure 2A must be improved or eliminated, the analysis of results are presented in figure 2b
Figure 5 The text must be adjusted to match the images
lines 159, 193 must be corrected, change included by induced
The text of section 3.4 (line 193 to 203) is not clear
Author Response
Dear Reviewer
Thanks for your comments concerning our manuscript entitled “Cytotoxicity and autophagy induced by Ivermectin via AMPK/mTOR signaling pathway in RAW264.7 cells” (Manuscript ID: molecules-2167456). Those comments are all valuable and helpful for revising and improving our paper. We have studied all comments carefully and have made conscientious correction. Revised portion are marker in red in paper. The main corrections in the paper and the responds to your comments are as following:
Comment 1: For example, in introduction or in discussion, it must be mentioned that this work is a continuation of their research, since the same group had already shown that ivermectin induces the death of RAW cells (Zhang P 2022 Chemosphere, 289), although in this article they do not yet describe the type of death.
Response: Thanks for your advice. We had added related content in introduction and discussion part.
Comment 2: It is necessary also to discuss that previously has been demostrated that ivermectin could induce autophagy in other cell line.
Response: Thanks for your advice. We had added related content in discussion part.
Comment 3: Another limitation of the work, that must be declared, is that RAW cells are tumor cells, and it has to be compared the effect in normal cells,even between resting macrophages, M1 and M2.
Response: Thank the review for pointing this out. In this study, we used RAW264.7 cells as a cell model to evaluate the autophagy effect of IVM in order to preliminarily explore the molecular mechanism of IVM immunotoxicity. Although RAW cells belong to mononuclear macrophages of mouse tumor line, they also have some physiological characteristics of normal mononuclear macrophages, which has certain scientific significance for the functional evaluation of immune system. Your suggestion, to use normal cells or pro-inflammatory M1 phase macrophages and anti-inflammatory M2 phase macrophages as cell models, is good and reasonable, and these cell lines will also be the most ideal cell models. However, in view of carefully evaluation of our experimental technique required to complete this supplementary work, we feel that we cannot afford this supplementary research at this time. And these supplementary experiment contents could be a valuable research direction in our follow-up research to explore.
Comment 4: Another limitation is the concentration of ivermectin tested, the minimun was 2.5 uM that is around 2 ug/mL.
Response: Thanks for your advice. The design of this concentration range refers to our previous research (Zhang P 2022 Chemosphere, 289).
Comment 5: By another hand, figure 2A must be improved or eliminated, the analysis of results are presented in figure 2b
Response: Thanks for your advice. The figure 2a has been removed, and the analysis of results in figure 2b was reserved.
Comment 6: Figure 5 The text must be adjusted to match the images
Response: Thanks for your advice. We have revised relevant text content about figure 5.
Comment 7: lines 159, 193 must be corrected, change included by induced
Response: We are very sorry for our careless mistake and they are rectified in relevant part.
Comment 8: The text of section 3.4 (line 193 to 203) is not clear
Response: Thanks for your question. The problem has been corrected in this section.

Reviewer 3 Report
The manuscript “Cytotoxicity and autophagy induced by Ivermectin via AMPK/mTOR signaling pathway in RAW264.7 cells” has been reviewed, please check comments below
1. Please mention brief methods and results used in the study
2. Line 27: cite reference
3. Line 37: “This study focused on the mechanism of autophagy” please revise or move it to discussion section. Focus in introduction in this section
4. Line 47: cite reference
5. Line: 89: Move to discussion section
6. 2.3: cite reference
7. 2.4: cite reference
8. 2.5: cite reference
9. 2.6: Please cite reference of method used
10. 2.7: cite reference
11. Line 196: remove “And”
12. Line: 233 cite reference
13. Include separate conclusion paragraph
Author Response
Dear Reviewer
Thanks for your comments concerning our manuscript entitled “Cytotoxicity and autophagy induced by Ivermectin via AMPK/mTOR signaling pathway in RAW264.7 cells” (Manuscript ID: molecules-2167456). Those comments are all valuable and helpful for revising and improving our paper. We have studied all comments carefully and have made conscientious correction. The main corrections in the paper and the responds to your comments are as following:
Comment 1: Please mention brief methods and results used in the study.
Response: Thanks for your advice. We had added relevant content in abstract part.
Comment 2: Line 27: cite reference
Response: Thanks for your advice. We had cited references in this part.
Comment 3: Line 37: “This study focused on the mechanism of autophagy” please revise or move it to discussion section. Focus in introduction in this section.
Response: Thanks for your advice. We had revised this part.
Comment 4: Line 47: cite reference
Response: Thanks for your advice. We had cited references in this part.
Comment 5: Line: 89: Move to discussion section
Response: Thanks for your advice. We had moved this sentence to discussion part.
Comment 6: 2.3: cite reference
Response: Thanks for your advice. We had revised this part.
Comment 7: 2.4: cite reference
Response: Thanks for your advice. We had cited reference in this part.
Comment 8: 2.5: cite reference
Response: Thanks for your advice. We had cited reference in this part.
Comment 9: 2.6: Please cite reference of method used
Response: Thanks for your advice. We had cited reference in this part.
Comment 10: 2.7: cite reference
Response: Thanks for your advice. We had cited reference in this part.
Comment 11: Line 196: remove “And”
Response: Thanks for your advice. We had revised this part.
Comment 12: Line: 233 cite reference
Response: Thanks for your advice. We had cited reference in this part.
Comment 13: Include separate conclusion paragraph
Response: Thanks for your advice. We had separated an independent conclusion paragraph.
